Traffic data imputation via knowledge graph-enhanced generative adversarial network

Liu Yinghui 1
Shen Guojiang 1
Liu Nali 1
Han Xiao 2
Xu Zhenhui 3
Zhou Junjie 3
http://orcid.org/0000-0003-2698-3319 Kong Xiangjie 1 xjkong@ieee.org
1 College of Computer Science and Technology, Zhejiang University of Technology , Hangzhou , China
2 School of Data Science, City University of Hong Kong , Hong Kong , China
3 Zhejiang Supcon Information Co., Ltd. , Hangzhou , China
Balas Valentina Emilia
Electronic publication date: 2024 Oct 14
Publication date: 2024
Volume: 10
Electronic Location ID: e2408
Received 2024 Jun 3; Accepted 2024 Sep 22
Copyright: © 2024 Liu et al.
Copyright year: 2024
Copyright holder: Liu et al.
License: This is an open access article distributed under the terms of the Creative Commons Attribution License, which permits unrestricted use, distribution, reproduction and adaptation in any medium and for any purpose provided that it is properly attributed. For attribution, the original author(s), title, publication source (PeerJ Computer Science) and either DOI or URL of the article must be cited.
License URL: https://creativecommons.org/licenses/by/4.0/

Keywords: Traffic data imputation, Generative adversarial networks, Knowledge graph

Funding: “Pioneer” and “Leading Goose” R&D Program of Zhejiang 2023C01241 National Natural Science Foundation of China 62072409 and 62073295 Zhejiang Provincial Natural Science Foundation LR21F020003 This work was supported by the “Pioneer” and “Leading Goose” R&D Program of Zhejiang 2023C01241, the National Natural Science Foundation of China under Grant 62072409 and Grant 62073295, and the Zhejiang Provincial Natural Science Foundation under Grant LR21F020003. There was no additional external funding received for this study. The funders had no role in study design, data collection and analysis, decision to publish, or preparation of the manuscript.

==============================
Traffic data imputation is crucial for the reliability and efficiency of intelligent transportation systems (ITSs), forming the foundation for downstream tasks like traffic prediction and management. However, existing deep learning-based imputation methods struggle with two significant challenges: poor performance under high missing data rates and the limited incorporation of external traffic-related factors. To address these challenges, we propose a novel knowledge graph-enhanced generative adversarial network (KG-GAN) for traffic data imputation. Our approach uniquely integrates external knowledge with traffic spatiotemporal dependencies to improve data imputation quality. Specifically, we construct a fine-grained knowledge graph (KG) that differentiates attributes and relationships of external factors such as points of interest (POI) and weather conditions, facilitating more robust knowledge representation learning. We then introduce a knowledge-aware embedding cell (EM-cell) that merges traffic data with these learned external representations, providing richer inputs for the spatiotemporal GAN. Extensive experiments on a large-scale real-world traffic dataset demonstrate that KG-GAN significantly outperforms state-of-the-art methods under various missing data scenarios. Additionally, ablation studies confirm the superior performance gained from incorporating external knowledge, underscoring the importance of this approach in addressing complex missing data patterns.

Introduction

Traffic detection data collected in intelligent transport systems (ITSs) often suffer from missing data due to various technical and management issues, including software failures, power outages, transmission errors, or storage failures (Tan et al., 2014), as shown in Fig. 1. For example, the Caltrans performance measurement system (PEMS) can be used to collect traffic data, calculate highway usage and congestion delays, predict travel time, evaluate ramp metering methods, and validate traffic theories. However, the data samples received by PEMS are often incomplete. According to the statistics, the ITS in Beijing, China was still under development in 2008, the daily traffic flow data had a general loss rate of around 10% (4% due to detector failure, 6% due to other reasons), and some loop detectors even generated a missing rate as high as 20–25% (Qu et al., 2009). Some extreme missing scenarios were reported in Alberta, Canada. Jianrui, Xingyi & Yi (2010) pointed out that nearly 50% of traffic data was missing in 7 years. Missing data seriously affects the accuracy and reliability of traffic forecasting (Olayode et al., 2024), management, and control systems (Kong et al., 2024; Pamuła, 2018). To address the missing data problem, data imputation is crucial to reconstruct the dataset by filling in the missing values with robust estimates.

Figure 1 Road network sensors with missing data.

Map data © 2024 Google.

Most previous imputation methods for traffic data fall into three categories: 1) Traditional statistical methods, such as support vector regression (SVR) (Wu, Ho & Lee, 2004), autoregressive integrated moving average model (ARIMA) (Cetin & Comert, 2006), mean imputation, median imputation (Kaiser, 2014) and other algorithms (Bania & Halder, 2020; Caillault, Lefebvre & Bigand, 2020). These methods rely on smoothness and periodicity to interpolate missing values. However, there is uncertain variation in real life, which leads to the unsatisfactory results of these methods; 2) Tensor decomposition methods (new machine learning-based methods) (Chen, He & Sun, 2019; Zhang et al., 2021; Chen et al., 2023, 2024). This category of methods estimates the missing values in the traffic flow by obtaining a suitable low-rank approximation of the incomplete matrix. 3) Deep learning methods (Kong et al., 2023; Ni & Cao, 2022; Shen et al., 2023; Tan et al., 2020; Tian et al., 2018). They interpolate data by learning the temporal and spatial correlation of data or the distribution of data.

However, deep learning methods achieve inferior performance compared to the tensor decomposition methods under high data missing rate. In addition, these methods are only limited to imputing data with traffic data itself. In addition to being influenced by the quality of road detectors and the distribution of spatiotemporal features, traffic information may also be affected by various external factors, which is the efficient promotion of knowledge-driven data imputation. For example, weather conditions, the existence of traffic stations, emergencies, holidays, and the distribution of nearby Points of Interest (POIs) (Lana et al., 2018; Liu et al., 2024; Xu et al., 2022). These external factors may affect urban traffic data directly or indirectly. For example, the traffic volume under different weather conditions may have different states as the weather changes over time. What’s more, traffic data is not only influenced by a single factor but also by various factors. For example, under the same heavy rain conditions, the traffic volume around schools is more affected than on less popular roads nearby. Integrating the semantic correlation of multi-source data information is the key to improving the ability to impute traffic data. Fortunately, knowledge graphs (KGs) (Li et al., 2023; Peng et al., 2023) that contain rich semantics about entities and relations provide a way to integrate different external factors and represent them in a unified manner.

In light of the above limitations and challenges, we propose a knowledge graph-enhanced generative adversarial network (KG-GAN) for spatiotemporal traffic data imputation. This approach is designed to address the complex nature of traffic data, which is influenced not only by temporal and spatial dependencies but also by various external factors such as points of interest (POI), weather conditions, and other contextual information. To tackle the challenge of effectively incorporating external factors, we use a traffic-specific KG construction approach that distinguishes the attributes and relationships of external entities. This allows the model to capture fine-grained semantic correlations that are often overlooked in traditional methods. The constructed KG serves as a rich source of prior knowledge, enhancing the representation learning process. After learning these knowledge representations, we introduce a knowledge-aware embedding cell (EM-cell). This component is specifically designed to seamlessly integrate the learned KG representations with traffic data, enriching the traffic embeddings with semantic information. These enriched embeddings are then fed into a spatiotemporal generative adversarial network, which is capable of generating high-quality imputed data by effectively modeling both the spatiotemporal dependencies and the complex external correlations. Compared to tensor decomposition methods that primarily focus on reducing the dimensionality of high-dimensional data, our KG-GAN approach benefits from the integration of prior knowledge in the form of KGs. This not only compensates for the limitations of purely data-driven deep learning methods but also enhances the model’s adaptability to complex and diverse missing data patterns, ultimately leading to more robust and accurate imputation results.

The following are our summarized contributions: To improve the performance of the deep learning imputation model while considering the complex influence of external factors on traffic interpolation, we propose a KG-enhanced approach (namely KG-GAN), in which traffic spatio-temporal characteristics and external knowledge graph are jointly learned.

We design a knowledge-aware embedding cell (EM-cell) to enrich model inputs by integrating traffic data with external knowledge, in which we construct implicit knowledge representations of external factors with a fine-grained KG construction approach that distinguishes the attributes and relations of external entities.

To demonstrate the effectiveness of KG-GAN, we conduct extensive experiments on a large-scale real-world traffic dataset showing that our method significantly outperforms existing imputation models and enhances performance across various missing data patterns. Further ablation experiments are performed to highlight the superiority of incorporating external knowledge learning.

The rest of this article is organized as follows. “Related Work” provides a systematic review of related works. “Methodology” describes the architecture and details of the proposed KG-GAN model. “Experiments” discusses the results of the experiment. Finally, the article is summarized in “Conclusion”.

Related work

Data imputation methods based on deep learning

Deep learning has been successfully applied in the field of data imputation. Che et al. (2018) proposed a deep model (GRU-D) based on learning gated recurrent units (GRU) (Cho et al., 2014). GRU-D employs two distinct representations of missing patterns, namely masking and time interval. Where the masking representation simulates the location of missing data, while the time interval representation represents the time range from the last observed value. GRU-D effectively integrates them into the deep model architecture. As a result, it can capture long-term temporal dependencies in time series. Furthermore, Cao et al. (2018) proposed a recurrent neural network (RNN) based model for the imputation of missing data in time series (BRITS). BRITS is used to interpolate missing values in time series data and can learn missing values directly in a bidirectional RNN without any specific preprocessing. It treats the interpolated values as variables of the RNN and can be efficiently updated during backpropagation. Both GRU-D and BRITS models are based on RNN, but they only consider the temporal correlation of data and do not consider the effect of spatial information on the imputed road network data. Generative adversarial networks (GANs) have been widely used in image processing (Xu et al., 2018; Yi, Walia & Babyn, 2019), and in recent years, they have been found to have good performance in data imputation. Yoon, Jordon & Schaar (2018) proposed a GAN-based GAIN model where they used generator and discriminator adversarial learning to model the distribution of the original data and then achieve the effect of imputing the missing data. Wang et al. (2021) proposed a PC-GAIN model which added a GAIN-based pre-training process. However, these GAN-based models do not take into account the spatial and temporal correlation of the data, resulting in unsatisfactory imputation of the traffic data. Ye, Zhang & Yu (2021) proposed a graph attention network model (GACN) for traffic missing data imputation, which follows an encoder-decoder structure and introduces a graph attention mechanism to learn the traffic graph. It allows higher-quality traffic data to be estimated by extracting typical spatiotemporal features. Different from previous works, we use a multi-perspective spatiotemporal generative adversarial network to analyze and extract traffic features from three perspectives: temporal, spatial, and spatiotemporal (Li et al., 2018; Shen et al., 2022).

Knowledge representation of traffic data

The generation of multi-source data is a natural consequence of a complex hybrid urban transport system. The relationships in multi-source data are mainly presented as networks, and mining the structural and relational information contained in the networks through representation vectors becomes the main method to capture the network information. In general, networks can be classified into homogeneous and heterogeneous networks based on the type of nodes. Most realistic traffic states are heterogeneous network structures, but the traditional HEBE (Gui et al., 2017) embedding framework for handling heterogeneous networks is only adapted to specific network architectures due to the limitation of meta-path accuracy. In recent years the application of knowledge graphs has gradually entered the public domain, and they are used in the traffic field for their excellent ability to handle graph structures and information (Muppalla et al., 2017; Xu et al., 2016). Typical knowledge graph representation learning methods include TransE (Bordes et al., 2013), TransH (Wang et al., 2014), and TransR (Lin et al., 2015). Compared to TransE and TransH, TransR constructs a projection matrix to model entities and relationships in both entity space and relationship space, and performs translation in the relationship space, breaking the limitation of the same space. Therefore, we choose TransR to model our constructed entities, relationships, and attributes.

Methodology

Overall framework

Our proposed KG-GAN model perceives the semantic information of the external knowledge graph and the spatiotemporal relevance of the traffic features through adversarial learning, thus effectively improving the accuracy of traffic data imputation. The model architecture diagram is shown in Fig. 2. The architecture takes in road network data, traffic speed data, and knowledge graphs to facilitate the learning of semantic knowledge and spatiotemporal dependencies, ultimately generating imputed data. In particular, following Lin, Liu & Sun (2016), we first divide the knowledge triad into a relation triad and an attribute triad to realize the refinement of the knowledge attribute and relationship. Then, we use the knowledge representation model with entities, attributes and relations (KR-EAR) to train the triad to generate the relational representation matrix to characterize the implicit knowledge. In addition, we propose a knowledge-aware embedding cell (EM-Cell) to fuse the implicit representation of the knowledge graph with traffic features for nonlinear self-learning, and input the learned-well traffic embedding with rich semantic information into our previously proposed multi-perspective spatiotemporal generative adversarial network (MST-GAN) (Shen et al., 2022) to guide the convergence and optimization of the model. Ultimately, it enables the model to interpolate data with complex correlations between traffic spatiotemporal features and external factors.

Figure 2 The KG-GAN model framework.

Given the incomplete traffic observed data X, the traffic imputation problem can be considered to learn an imputation function Func, which can calculate an appropriate value for each missing component in X based on the traffic network structure matrix A and the knowledge graph (KG) as follows:

(1) Ximputed=Func(A,X,KG).

Combined with the adversarial training of the MST-GAN model, the final min-max objective for the overall model optimization is:

(2) minGT,GS⁡maxD⁡EX¯,M[M⋅log⁡D(X¯)+(1−M)⋅log⁡(1−D(X¯)],

where GT characterizes the temporal generator, GS characterizes the spatial generator, D characterizes the discriminator, X¯ denotes the road network data simulated by the temporal and spatial generator, M characterizes the masking matrix, and log⁡(⋅) characterizes the logarithmic calculation of the elements.

The design and learning of knowledge graph

KG is a semantic network-based knowledge base that uses a directed graph structure to organize data such as entities, relationships, and attributes. The advantages of KG, such as their ability to integrate diverse information sources and preserve both semantic and structural relationships, are particularly beneficial for traffic data imputation. In handling incomplete traffic data, KG can effectively model the complex, multi-relational dependencies inherent in traffic networks. By representing heterogeneous nodes and multi-relationship information, KG can construct hierarchical and semantic relationships between various traffic-related entities (e.g., road segments, sensors, events) (Ning et al., 2024). This hierarchical and semantic structuring allows for more accurate imputation by leveraging the rich contextual and relational information within the graph, which is critical in capturing the dynamic and interdependent nature of traffic systems.

Distributed knowledge representation (KR) encodes entities and relations in a low-dimensional semantic space, significantly improving the performance of relation extraction and knowledge inference. In many KGs, some relations represent attributes of entities (properties), while others represent relationships between entities (relations). Traditional KR methods treat all relations equally and usually have poor accuracy in modeling one-to-many and many-to-one relations (consisting mainly of attributes). In principle, a knowledge graph representation that distinguishes between attribute and relationship information is more suitable for capturing semantic information and relevance in this context. Therefore, we use the knowledge graph representation method knowledge representation learning with entities, attributes and relations (KR-EAR) (Lin, Liu & Sun, 2016) based on entity-attribute and entity-relationship to capture knowledge structure and semantic information between road parts and external factors. The KR-EAR and the traditional KR method are shown specifically in Fig. 3, where A1 and A2 are the two attributes. The value set of attribute A1(V1) contains e6 and e7 which are squares (also colored in blue), while A2(V2) contains e8 and e9 which are hexagonal (also colored in grey). In the traditional KR representation method (left), attributes A1 and A2 are treated as relations ra and rb. In contrast, KR-EAR encodes the relational triples using the traditional KR representation method and treats attribute prediction as a classification problem.

Figure 3 The KR-EAR (right) and traditional KR method (left).

In this article, roads, attributes, and the relationships between them are represented as a triad of KG = {R,ATT, Relations}. Specifically, the triads are divided into three categories:

1) Road adjacency triple R (head entity, relationship, tail entity)

(3) R={(vi,adj,vj)},i,j∈{1,2,...,n},

where R is a relational triplet representing the adjacency relationship adj between segments vi and vj, and n is the number of segments.

2) Attribute triple ATT (entity, attribute, attribute value)

(4) ATT={(vi,al,aval)},l∈{1,2,...,L},

where al is the l-th class of attributes, aval is the corresponding attribute value (e.g., weather overcast), and L is the number of attribute classes.

3) Attribute co-occurrence triple Relations (attribute 1, attribute 2, co-occurrence probability)

(5) Relations={(al1,al2,p)},l1,l2∈{1,2,...,L},

where al1 and al2 denote two different attributes of an entity, p is their co-occurrence probability and the attribute co-occurrence probability describes the probability that two attributes exist in the same section.

Given a KG, the objective of KR-EAR is to learn the representations XE of entities, relations, and attributes. The objective function is defined as maximizing the joint conditional probability of the relationship triple and the attribute triple, which is formalized as:

(6) P(R,ATT|XE)=P(R|XE)P(ATT|XE),=∏(vi,adj,vj)∈R⁡P((vi,adj,vj)|XE)∏(vi,al,aval)∈ATT⁡P((vi,al,aval)|XE),

where P((vi,adj,vj)|XE) denotes the conditional probability of the relation triple (vi,adj,vj) and P((vi,al,aval)|XE) is the conditional probability of the attribute triple (vi,al,aval). P((vi,adj,vj)|XE) is generated by an energy function e following TransR (Lin et al., 2015):

(7) P((vi,adj,vj)|XE)=exp(e(vi,adj,vj))∑v^i∈Vexp(e(vi^,adj,vj)),e(h,r,t)=−||hMr+r−tMr||L1/L2+br,

where Mr denote the projection matrix which may projects entities from entity space to relation space. br is a bias constant and V is a set of road section entities. P((vi,al,aval)|XE) is captured by a scoring function (Lin, Liu & Sun, 2016):

(8) P((vi,al,aval)|XE)=exp(s(vi,al,aval))∑av^al∈AValexp(s(vi,al,av^al)),

where s() is the scoring function for each attribute value of a given entity and AVal is the attribute value set. In this way, KR-EAR generates representations of relations and attributes while strengthening the correlations between attributes.

The fusion of knowledge representations and GAN

In order to better model the spatiotemporal dependencies of traffic data and perceive the influences of external factors from multiple perspectives, as well as the correlations between factors, this study proposes a knowledge-aware embedding method, namely EM-Cell. The design of EM-Cell is based on a deep analysis of traffic data and the mining of multi-source knowledge relationships, which can effectively fuse the complex knowledge representations of spatiotemporal changes of traffic data and external factors. The details of EM-Cell are shown in Fig. 4, where the input consists of two parts: the knowledge representation matrix XE constructed by KR-EAR and the road segment feature Xt observed at time t. Due to the diversity of external factors, this article divides them into two categories: static factors and dynamic factors. Specifically, Es and Ed in Fig. 4 represent the embeddings of road segments with respect to static external factors (such as shopping mall information, hospital information) and dynamic external factors (such as weather changes), respectively, processed by KR-EAR. Therefore, the fusion operation formula between the traffic feature matrix and the knowledge representation matrix designed in this study is as follows:

(9) Xt′=Concat[σ(fs(Es,Xt)),σ(fd(Ed,Xt))],wheref(x,y)=xyW+b,f={fs,fd}.

Figure 4 Structure of EM-cell.

Both W={Ws,Wd} and b={bs,bd} denote the learnable parameters. σ is the sigmoid function.

To model the spatiotemporal dependence of traffic data based on knowledge representation, we use the updated road segment features Xt′ and the adjacency matrix A as the input to the spatiotemporal generative adversarial network. We use the MST-GAN model for data imputation because it considers advanced multi-view spatiotemporal fusion through chain generator adversarial learning. To achieve multi-view feature fusion, MST-GAN uses an adversarial between a chain generator and a discriminator to achieve a high-level fusion of temporal and spatial information. The generator learns different enhanced features flexibly at different stages using independent parameters. In summary, MST-GAN captures the temporal and spatial correlation of traffic data through a bidirectional recurrent network and a graph convolutional network. In addition, it introduces an attention layer to compute dynamic weights between different time points to focus on key temporal features.

Specifically, the temporal generator GT uses a bi-directional long and short-term memory network based on the attention mechanism ( BiLSTM_ATT) as the kernel, and the final output value of the temporal generator X^ is denoted as:

(10) XBiLSTM_ATT=Attention(BiLSTM(Xt′)),

(11) X^=Xt′⊙M+XBiLSTM_ATT⊙(1−M),

where ⊙ means multiplying by elements and M represents the mask matrix. The spatial generator GS kernel consists of a graph convolutional network (GCN), containing two convolutional layers and a fully connected layer. As the depth of the network increases, it brings many problems such as gradient dissipation. Therefore, the model uses skip connect to improve the gradient dissipation problem during backpropagation. The output of the spatial generator is expressed as:

(12) X¯=Xt′⊙M+GS(X^)⊙(1−M).

Both time and spatial generators use MSE as the loss function and Adam as the optimization function, the loss gradually decreases to stability during the cyclic iteration of the model. Eventually it generates interpolated data with higher quality.

The discriminator D is used to distinguish the data as true or false. The generator tries hard to make the simulated data closer to the true value, while the discriminator tries hard to identify the data as true or false. The core structure of the discriminator network consists of GCN and BiLSTM_ATT. The loss function of the discriminator can be expressed as:

(13) LD=−1n∑i=1n(M⋅log⁡D(X¯)+(1−M)⋅log⁡(1−D(X¯)),

where n is the sample number.

KG-GAN training for missing data imputation

The detailed training procedure of KG-GAN is presented in Algorithm 1. Firstly, refine entity attributes and entity relationships using the KR-EAR method to construct triples of correlated knowledge and derive knowledge representation matrix XE. Then, through the EM-Cell module, fuse the spatiotemporal traffic data Xt (which may contain missing data) with the dynamic and static matrices of the knowledge representation matrix to obtain integral embedding of rich traffic information. Finally, at each training time step, the fused input Xt′ is subjected to adversarial optimization in the MST-GAN network. During the training phase of the MST-GAN model, the discriminator is pre-trained to learn the characteristics of the generated data and observed data. Then, the discriminator and two generators are trained adversarially. In detail, the MST-GAN model uses the temporal generator to train hyperparameter θGT from a temporal perspective. Next, we use the spatial generator to train hyperparameter θGS from a spatial perspective. The learning of θGT and θGS allows us to refine the extraction of spatiotemporal features in stages.

Algorithm 1 KG-GAN model training for data imputation.

Input: Original complete traffic dataset Xt(m×n); road adjacency triplet R; attribute triplet ATT; attribute co-occurrence triplet Relations; loss hyperparameters α; masking matrix M; indicator matrix H; the number of epochs N and the initialized parameters: generator θGT, generator θGS, and discriminator θD	
  1: Construct a KG = {R, ATT, Relations} triplet.	
  2: By maximizing the P(R,ATT|XE) obtain the knowledge representation matrix XE.	
  3: Integrating Xt and XE : Xt′=EM- Cell(Xt,XE)	
  4: for epoch=1,2,…,N	
  5:    (1) Discriminator optimization:	
  6:       Obtain discriminator loss LD via Eqs. (10–13)	
  7:       Back-propagate LD to update θD	
  8:    (2) Generator optimization:	
  9:       Obtain the output of GTX^ via Eqs. (10), (11)	
 10:        LR1=1n∑i=1n||(Xt′−X^)⊙M||22	
 11:        LGT←−D(X^,H)+αLR1	
 12:       Obtain the output of GSX¯ via Eq. (12)	
 13:        LR2=1n∑i=1n||(Xt′−X¯)⊙M||22	
 14:        LGS←−D(X¯,H)+αLR2	
 15:       Back _propagate LGT, LGS to update θGT, θGS	
 16: endfor	
 17: Impute the missing values:	
 18: Obtain imputed data Ximputed via Eqs. (10–12)	
Output: Trained parameters θGT, θGS, and θD; imputed data Ximputed	

Experiments

Dataset

The dataset contains road network data, weather data, POI data and traffic speed data for each street in Luohu District, Shenzhen, where the time span is from January 1, 2015 to January 31, 2015 (Zhu et al., 2022). Weather data is divided into five categories: sunny, light rain, heavy rain, cloudy and foggy. POI data is divided into nine categories: business, transportation, medical, living, accommodation, education, food, shopping and others. Due to the limitations of experimental data collection, it is difficult to have existing publicly available datasets that collect traffic speed data, road network data, and external correlates (Weather, POI, etc.,) at the same time, so only one regional dataset from Shenzhen is used in this experiment. Nevertheless, as our model is universal and transferable, researchers can validate it in any city that gives a relevant dataset.

Knowledge representation

The inputs to the knowledge representation model include attribute triple, road adjacency triple, and attribute co-occurrence triple. Based on the composition of the Shenzhen dataset, we construct POI attribute triples using road section number, POI category, and numbers of POI. We also use time, weather, and relevance to construct weather attribute triples, resulting in the construction of a knowledge graph of the Luohu District in Shenzhen. Specific examples are as follows: (road 123, enterprise, 3) indicates that there are three enterprises located around road section 123; (road 100, hospital, 1) indicates that a hospital exists around road section 100; (road 100, weather conditions, moment t) and (moment t, weather, clear) indicate that road 100 has clear weather condition at moment t.

Experimental design

In this article, 80% of the dataset is used as training data and the rest of the data was used as test data. We choose T = 288 time steps (i.e., 15 min × 288 = 72 h) as the imputation window. During training, we use the sliding window method to impute [ t, t + T], [ t + T, t + 2T], [ t + 2T, t + 3T], etc. We initialize all weight values uniformly and normalize the input data to [0,1]. Both GCN and BiLSTM_ATT networks contain two layers whose sizes are 128. The model is trained using the Adam optimizer with an initial learning rate of 0.01. As displayed in Fig. 5, we validate the performance of the model under three missing modes (Li et al., 2018; Liang, Zhao & Sun, 2021): 1) Random missing (RM), where missing values are completely independent of each other and displayed as randomly scattered points for each sensor (or road); 2) Temporal correlated missing (TCM), where missing values are dependent in the time dimension and appear as a consecutive time interval for each sensor (or road); 3) Spatially correlated missing (SCM), where missing values are dependent in the spatial dimension and appear at neighboring sensors or connected road links for each time slot. The performance of the model is compared with other baseline methods at different deletion rates from 30% to 80%. The model proposed in this study has an important hyperparameter, namely the knowledge embedding dimension, which has a significant impact on the data imputation results. After conducting multiple experiments, TransE and TransR models are used to learn the knowledge graph, and their performances are compared using embedding dimensions of 20 and 15 for TransE, and 15, 20, and 50 for TransR, as shown in Table 1. The comparison is performed in the typical scenario of 50% missing data, and the RMSE and MAE loss metrics are used to evaluate the models with different embedding dimensions. Based on the results, an embedding dimension of 15 is chosen to achieve the best final imputation results. Fig. 6 shows the traffic speed data imputation results under a missing rate of 50%, a knowledge embedding dimension of 15, a random missing pattern, and an RMSE evaluation metric.

Figure 5 Patterns of missing data.

Table 1 Embedded dimension selection effect value.

Dimensions	Missing patterns	
RM	TCM	RCM	
MAE	RMSE	MAE	RMSE	MAE	RMSE	
TransE(20)	0.0265	0.0485	0.0284	0.0477	0.0278	0.0509	
TransE(50)	0.0260	0.0461	0.0275	0.0507	0.0274	0.0546	
TransR(15)	0.0270	0.0454	0.0264	0.0445	0.0265	0.0457	
TransR(20)	0.0370	0.0463	0.0265	0.0505	0.0325	0.0521	
TransR(50)	0.0279	0.0613	0.0333	0.0521	0.0269	0.0556	
Note:

Values in bold indicate the best result.

Figure 6 The visualization of data imputation results.

Baselines

To demonstrate the effectiveness of our model in all aspects, we compared several baseline experiments. These include the machine learning methods: MEAN and SVR (Wu, Ho & Lee, 2004); the vector decomposition method: BGCP (Chen, He & Sun, 2019); the deep learning methods: GRU-D (Che et al., 2018), BRITS (Cao et al., 2018), GAIN (Yoon, Jordon & Schaar, 2018), PC-GAIN (Wang et al., 2021), GACN (Ye, Zhang & Yu, 2021), DGCRIN (Kong et al., 2023). MEAN: The missing elements are interpolated with the means of all relevant features.

SVR: We choose support vector machines as the representative of regression-based machine learning imputation methods.

GRU-D: GRU-D is a deep learning model architecture represented by two missing modes of masking and time interval, which improves model imputation performance through the application of the decay mechanism.

BRITS: BRITS is a missing value imputation method for time series data based on RNN, which can directly learn missing values in a bidirectional recursive dynamical system without the need for any specific assumptions.

GAIN: GAIN is a GAN-based method for unsupervised missing data imputation. It adds an indicator matrix to the GAN, which is to ensure that the generator generates samples according to the true underlying data distribution.

PC-GAIN: PC-GAIN is a method of unsupervised missing data imputation. It proposes a kind of potential category information contained in a subset of low-missing rate data during pre-training while using synthetic pseudo-labels to identify auxiliary classifiers and then combines classifiers into GAN to help generators produce higher-quality prediction results.

BGCP: BGCP extends the Bayesian probability matrix decomposition model to higher order tensor and applies it to the task of spatiotemporal traffic data. They focus not only on the configuration of the model, but also on the data representation (i.e., matrix, third-order and fourth-order tensor).

GACN: GACN is a graph attention convolutional network model for missing data imputation, which follows an encoder-decoder structure. As a typical spatiotemporal imputation model, GACN introduces a graph attention mechanism to learn the spatial correlation of adjacent sensors. In addition, it superimposes temporal convolutional layers to extract relationships in time series.

DGCRIN: DGCRIN is also an imputation model based on dynamic graph convolutional networks and realizes state-of-the-art performance. It develops a graph generator to model dynamic spatial correlations and uses a dynamic graph convolutional gated recurrent unit to capture spatiotemporal relevances.

Evaluation metrics. We choose mean absolute error (MAE), root mean square error (RMSE) and mean absolute percentage error (MAPE) as evaluation indicators. Here’s how the evaluation metrics are calculated:

(14) MAE=1n∑i=1n||y^i−yi||

(15) RMSE=1n∑i=1n(y^i−yi)2

(16) MAPE=1n∑i=1n|y^i−yiyi|×100%

where n represents the number of missing data, yi^ represents the prediction of missing value, and yi represents the observed value.

Comparison experiment

Table 2 shows the error values of the KG-GAN and several baseline models under different missing rates and evaluation metrics. The data is the statistical result under RM missing pattern and we draw the following analyses: (1) Traditional imputation methods MEAN and SVR perform poorly compared to most deep learning methods, especially when the missing rate is higher than 50%; (2) compared to the traditional methods, RNN-based methods GRU-D and BRITS consider the time series correlation of data and have smaller imputation errors. However, the imputation performance of GRU-D is slightly lower than that of the BRITS algorithm, which is probably because GRU-D is more suitable for imputing medical data than traffic data; (3) GAIN and PC-GAIN are data imputation models based on data distribution. These GAN-based methods only adapt to low missing rate situations from the perspective of data distribution; (4) even at higher missing rates, the GACN model achieves a MAE of less than 6% and a RMSE of under 8%. Meanwhile, DGCRIN demonstrates even better performance. This highlights the significance of incorporating spatiotemporal correlations for traffic data imputation, which outperforms algorithms that only take temporal correlations and data distribution into account. (5) The MAE, RMSE and MAPE of KG-GAN are lower than the best baseline DGCRIN and superior to other baselines, which fully illustrates the effectiveness of our model.

Table 2 Experimental results of a data imputation method in RM pattern.

Models	Missing rate	
30%	40%	50%	60%	70%	80%	
MAE	RMSE	MAPE	MAE	RMSE	MAPE	MAE	RMSE	MAPE	MAE	RMSE	MAPE	MAE	RMSE	MAPE	MAE	RMSE	MAPE	
MEAN	0.0321	0.0868	2.16%	0.0476	0.1100	3.28%	0.0671	0.1361	5.16%	0.0911	0.1650	7.86%	0.1206	0.1973	9.45%	0.1540	0.2306	10.98%	
SVR	0.0761	0.0880	5.67%	0.0783	0.0909	5.95%	0.0789	0.0921	6.13%	0.0831	0.0965	6.87%	0.0874	0.1035	7.03%	0.0951	0.1108	7.84%	
GRU-D	0.0763	0.0944	5.88%	0.0822	0.1010	6.07%	0.0908	0.1113	7.16%	0.0917	0.1120	7.05%	0.1009	0.1222	8.20%	0.1044	0.1263	8.39%	
BRITS	0.0526	0.0794	3.00%	0.0550	0.0818	3.73%	0.0580	0.0849	4.07%	0.0603	0.0867	4.54%	0.0624	0.0884	4.22%	0.0660	0.0923	4.51%	
GAIN	0.0361	0.0547	2.07%	0.0399	0.0613	2.19%	0.0491	0.0734	2.84%	0.0617	0.1088	4.97%	0.0948	0.1716	8.14%	0.1173	0.1881	9.48%	
PC-GAIN	0.0841	0.1417	6.85%	0.0875	0.1474	7.33%	0.0926	0.1524	8.21%	0.0955	0.1543	8.59%	0.1000	0.1582	9.13%	0.1080	0.1670	10.25%	
BGCP	0.0573	0.0794	3.88%	0.0572	0.0794	3.97%	0.0574	0.0795	4.10%	0.0580	0.0801	4.54%	0.0584	0.0805	4.95%	0.0592	0.0817	5.05%	
GACN	0.0534	0.0768	3.26%	0.0549	0.0771	3.39%	0.0557	0.0775	3.77%	0.0562	0.0782	3.56%	0.0575	0.0789	4.17%	0.0584	0.0800	4.35%	
DGCRIN	0.0463	0.0711	1.75%	0.0469	0.0715	2.47%	0.0480	0.0723	2.61%	0.0503	0.0746	2.89%	0.0517	0.0757	3.59%	0.0523	0.0761	3.35%	
KG-GAN	0.0244	0.0427	1.44%	0.0254	0.0444	1.77%	0.0270	0.0454	2.22%	0.0291	0.0488	2.05%	0.0292	0.0523	2.58%	0.0309	0.0535	2.85%	
Note:

Values in bold indicate the best result.

From the perspective of robustness analysis, Fig. 7 shows the performance comparison of various baseline models under SCM and TCM. The horizontal axis in the figure represents different data missing rates, and the vertical axis is used to visualize the loss results of different models under different evaluation metrics. Based on the experimental results, the following conclusions can be drawn: (1) The traditional MEAN method is basically unaffected by the missing pattern, and the loss value steadily increases with the increase of the missing rate in different evaluation metrics. When the missing rate is greater than 60%, its RMSE metric is higher than all other baseline models, indicating poor robustness; (2) the robustness of the SVR algorithm is superior to the MEAN method, and its performance under SCM is better than under TCM; (3) the GRU-D model mainly focuses on capturing time dependencies rather than spatial dependencies, so its performance under the time-continuous missing pattern is lower than under the space-continuous missing pattern. Moreover, as the missing rate increases, the loss value of GRU-D continues to rise, indicating that the robustness of the GRU-D model for different missing patterns and missing rates is not ideal; (4) intuitively, it can be found that the robustness of the GAIN model is only better than the traditional MEAN method. When the missing rate is greater than 40%, the loss curve of GAIN suddenly rises. This is because when the missing rate gradually increases, it is difficult for GAN to learn from historical data; (5) under different missing patterns and evaluation metrics, the robustness of BGCP, BRITS, GACN, DGCRIN and our KG-GAN are all very superior.

Figure 7 (A–F) Performance comparison in two missing patterns.

In general, since KG-GAN not only considers data distribution and spatiotemporal correlation dependencies but also introduces knowledge embedding of external related factors, the imputation performance of our model is the best.

Ablation experiments

To verify the gain effect of knowledge representation on the performance of generating adversarial network imputation data, this study compares KG-GAN with the MST-GAN model (i.e., KG-GAN w/o EM-Cell), which differs in that KG-GAN introduces a knowledge learning module to model and process knowledge representation of multi-source traffic information. This study conducted ablation experiments under SCM and TCM in terms of MAE, RMSE and MAPE as shown in Fig. 8. The results intuitively suggest that as the rate of missing data increases, the accuracy of both models decreases. This is attributed to the fact that the increasing amount of missing data negatively impacts the models’ learning ability. The study also indicates that in the case of missing data, it is necessary to use models with better robustness to handle data, and KG-GAN’s robustness is significantly better than MST-GAN’s. In terms of model accuracy, KG-GAN’s three indicators are better than MST-GAN’s, especially in the MAE evaluation indicator, highlighting KG-GAN’s superior imputation performance.

Figure 8 (A–F) Ablation experiment in SCM and TCM missing patterns.

In addition, in order to verify that the convergence speed of the KG-GAN is better than that of the MST-GAN, an ablation experiment is also designed to measure the trend of the loss under RM missing pattern, with a data missing rate of 50% and a knowledge embedding dimension of 15, as shown in Fig. 9. The horizontal axis represents the training batch (epoch) of the model, and every 100 epochs takes an average of 5 s. One can see that the KG-GAN model converges faster than the MST-GAN model since the KG-GAN model learns the spatiotemporal features of the dataset faster during training, and can better capture the missing patterns in the data.

Figure 9 Convergence rate in RM missing patterns.

In summary, the results of the ablation experiments prove that external factors (such as weather, POI, etc.) and the complex correlations between external factors are of great significance to the imputation of traffic data. By using knowledge graphs as prior knowledge to guide the training of the model, the convergence speed, efficiency, and accuracy of the model can be improved, ultimately achieving high-quality imputation of traffic data.

Conclusion

In this article, we propose KG-GAN, a knowledge graph-enhanced model for spatiotemporal traffic data imputation, designed to improve deep learning-based imputation models’ performance under high data scarcity and account for complex external factors. Our extensive experiments on a real-world traffic dataset validate KG-GAN’s effectiveness in traffic data imputation. Specifically, we first construct an implicit knowledge representation of external factors using a fine-grained knowledge graph, which accurately distinguishes between attributes and relationships. Next, we introduce a knowledge-aware embedding cell that integrates traffic data with the external knowledge representation, resulting in a refined traffic embedding. This embedding is then input into the MST-GAN model, facilitating effective convergence to the real traffic data distribution. Our approach achieves superior results by combining spatiotemporal feature learning with external knowledge, leading to more accurate imputation. The implications of our research are significant for traffic data imputation and intelligent transportation systems. By addressing data scarcity and incorporating external factors, our model offers a more robust solution for real-world applications, potentially improving traffic management and decision-making. Future work will focus on refining the knowledge graph to reduce noise, such as irrelevant connections between items and entities, which can impact model performance (Yang et al., 2023). We also plan to explore the application of KG-GAN to other domains and datasets to further validate its generalizability and robustness.

Supplemental Information

Supplemental Information 1 Containing the code for KG-GAN as well as the dataset for the experiments.

Supplemental Information 2 The graph data from Shenzhen.

Supplemental Information 3 The knowledge graph data from Shenzhen.

Supplemental Information 4 The poi embedding from Shenzhen.

Supplemental Information 5 The poi data from Shenzhen.

Supplemental Information 6 The traffic speed data from Shenzhen.

Supplemental Information 7 The weather data from Shenzhen.

Additional Information and Declarations

Competing Interests

Author Contributions

Data Availability

Xiangjie Kong is an Academic Editor for PeerJ. Zhenhui Xu and Junjie Zhou are employed by Zhejiang Supcon Information Co., Ltd.

Yinghui Liu conceived and designed the experiments, performed the experiments, analyzed the data, performed the computation work, prepared figures and/or tables, authored or reviewed drafts of the article, and approved the final draft.

Guojiang Shen conceived and designed the experiments, analyzed the data, performed the computation work, authored or reviewed drafts of the article, and approved the final draft.

Nali Liu conceived and designed the experiments, performed the experiments, analyzed the data, performed the computation work, authored or reviewed drafts of the article, and approved the final draft.

Xiao Han conceived and designed the experiments, performed the experiments, analyzed the data, performed the computation work, authored or reviewed drafts of the article, and approved the final draft.

Zhenhui Xu conceived and designed the experiments, prepared figures and/or tables, authored or reviewed drafts of the article, and approved the final draft.

Junjie Zhou conceived and designed the experiments, prepared figures and/or tables, authored or reviewed drafts of the article, and approved the final draft.

Xiangjie Kong conceived and designed the experiments, prepared figures and/or tables, authored or reviewed drafts of the article, and approved the final draft.

The following information was supplied regarding data availability:

The code and raw data are available in the Supplemental Files.

These data are from the original source at GitHub: https://github.com/lehaifeng/T-GCN/tree/master/KST-GCN.

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
