# Peer review of "Traffic data imputation via knowledge graph-enhanced generative adversarial network"

_PeerJ Computer Science, doi:10.7717/peerj-cs.2408_

## Round 0.1 · original submission · Major Revisions

The authors must improve the paper according to reviewers observations

Reviewer 1 ·

Basic reporting

This submission addressed the problem of traffic data imputation, which is of great significance in ITS. The authors develop a knowledge graph enhanced GAN to account for external factors that are not sufficiently modeled in prior studies. In general, the paper is well-written and –organized. Some comments are as follows:
1) The literature review should include more recent papers that are published in 2023 and 2024;
2) According to Figure 4, the traffic speed Xt’ is obtained by integrating the impacts of both static and dynamic factors. I wonder if the concatenation in Eq.(9) can capture the spatio-temporal dynamics of these impacts. Please justify.

Experimental design

1) The proposed method is quite simple and logically sound. But the introduction of the method needs more details and justification. Please elaborate.
2) Please report MAPE in your experimental results;
3) The study region and road network can be illustrated by a map.

Validity of the findings

No comment.

Reviewer 2 ·

Basic reporting

This paper proposes a Knowledge Graph-enhanced Generative Adversarial Network (KG-GAN) for traffic data imputation, in which we consider external knowledge while compensating for the limitation of deep learning imputation methods. Some problems should be addressed:
1. Many definitions of letters in equations are absent. Also, some definitions should be given completely, e.g. matrix X. How to define function Func? TransR()?
2. When introducing new modules in the proposed method, the motivations should be better clarified. For example, the authors devote to describe the advantages of Knowledge Graph (KG) in “The Design and Learning of Knowledge Graphs”. The authors should focus more on addressing why the advantages can handle the problem of traffic data imputation.
3. Solving missing data problem in ntelligent Transport Systems (ITSs) is a difficult problem, there have been a lot of works present, e.g, Composite Nonconvex Low-Rank Tensor Completion With Joint Structural Regression for Traffic Sensor Networks Data Recovery," IEEE Transactions on Computational Social Systems, 2024; A Novel Nonconvex Low-rank Tensor Completion Approach for Traffic Sensor Data Recovery from Incomplete Measurements,” IEEE Transactions on Instrumentation & Measurement, 2023. Also

Experimental design

no comment

Validity of the findings

no comment

Additional comments

no comment

Reviewer 3 ·

Basic reporting

No comment

Experimental design

No comment

Validity of the findings

No comment

Additional comments

Traffic data imputation via knowledge graph-enhanced generative adversarial network
Abstract
1. I am not convinced by the whole abstract; the abstract is more descriptive and should be rewritten or restructured. The first two lines “Traffic data imputation is an important component of intelligent transportation systems (ITSs), laying the foundation for downstream traffic tasks” is not connecting with the next sentence.
2. The first lines of the abstract are all about the statement of the problem and the significance of the research, the current abstract deviates from the structure of a traditional abstract.
3. Another major flaw with the abstract is that the methodology and the results is missing.
4. The methodology section in the abstract is weak. Where is the methodology, the breakdown of how the data was used?
5. Conclusively, the abstract needs to be rewritten. There are so many confusing sentences and sentences that do not give clarity.

Introduction
1. The third contribution of this research is really not a contribution, because based on this research, the authors are expected to validate if their results outperform other models.
2. The introduction section is also too brief, and the authors should add more in terms of problem statement and why conducting this research is important.
3. The authors should cite the following articles to buttress further their introduction and literature review section; https://doi.org/10.1016/j.ins.2023.01.113 and https://doi.org/10.1016/j.ijtst.2023.04.004

4. The key contribution should be combined to two and not three
5. I will recommend that the authors look throughout the introduction section and cite references that need to be cited because some statements in the introduction section need references.
6. The authors need to employ the services of an English editor to check the grammatical errors in this manuscript.
7. All acronyms should be properly defined by introducing a table.
8. Lines 95 and lines 120 are unnecessary.
9. The authors should go into more detail with Figure 2 and also try to enlarge the figure.


Methodology
1. The authors should include a flowchart of the methodological steps used in this study.
2. Can the authors show a graphical image of the streets of Luohou District Shenzhen
3. Why focus on the data from 2015?
Conclusions
1. The conclusion needs to be rewritten. The authors need to explain how their results were able to achieve their objectives and the implication of their research in the field of traffic data imputation and intelligent transportation system
2. Why is their model better than other models?
3. What is the limitation of this research and future recommendations?

---

## Round 0.2 · accepted · Accept

The paper was very well improved. It can be accepted.

Reviewer 3 ·

Basic reporting

I am satisfied with the response to reviewer's comments

Experimental design

I am satisfied with the response to reviewer's comments

Validity of the findings

I am satisfied with the response to reviewer's comments

Additional comments

I am satisfied with the response to reviewer's comments